# Progesterone Receptor Expression Level Predicts Prognosis of Estrogen Receptor-Positive/HER2-Negative Young Breast Cancer: A Single-Center Prospective Cohort Study

**DOI:** 10.3390/cancers15133435

**Published:** 2023-06-30

**Authors:** Youngji Kwak, Sung Yoon Jang, Joon Young Choi, Hyunjun Lee, Dong Seung Shin, Yeon Hee Park, Ji-Yeon Kim, Jin-Seok Ahn, Byung Joo Chae, Jonghan Yu, Jeong Eon Lee, Seok Won Kim, Seok Jin Nam, Jai Min Ryu

**Affiliations:** 1Division of Breast Surgery, Department of Surgery, Samsung Medical Center, School of Medicine, Sungkyunkwan University, Seoul 06351, Republic of Korea; youngji.kwak@samsung.com (Y.K.); sung.y.jang@samsung.com (S.Y.J.); joonyoung2.choi@samsung.com (J.Y.C.); hyunjun0403.lee@samsung.com (H.L.); dongseung.shin@samsung.com (D.S.S.); bj.chae@samsung.com (B.J.C.); jonghan.yu@samsung.com (J.Y.); jeongeon.lee@samsung.com (J.E.L.); seokwon1.kim@samsung.com (S.W.K.); seokjin.nam@samsung.com (S.J.N.); 2Division of Hematology-Oncology, Department of Medicine, Samsung Medical Center, School of Medicine, Sungkyunkwan University, Seoul 06351, Republic of Korea; yeonh.park@samsung.com (Y.H.P.); jyeon25.kim@samsung.com (J.-Y.K.); jinseok.ahn@samsung.com (J.-S.A.)

**Keywords:** breast neoplasm, progesterone receptor, prognosis

## Abstract

**Simple Summary:**

Although ER expression levels affect the prognosis of breast cancer, studies about PR expression levels are insufficient. Furthermore, there is a knowledge gap between single HR-positive and double HR-positive, especially according to PR expression. As HR positivity is an important prognostic factor, particularly in YBC patients, this research was conducted in a prospective cohort with only YBC patients in order to find out whether the expression of PR modifies the clinical course of breast cancer. We investigated clinicopathologic features and prognosis of ER-positive/HER2-negative breast cancer after stratifying them according to PR expression levels. Conclusively, low PR expression was correlated with worse clinicopathologic characteristics, and associated with increased risk of recurrence, distant metastasis, and death compared with strong PR expression group. Low PR might be a prognostic factor of ER-positive/HER2-negative YBC.

**Abstract:**

Background: Although estrogen receptor (ER) expression levels affect the prognosis of breast cancer, studies about progesterone receptor (PR) expression levels are insufficient, especially in young breast cancer (YBC). The purpose of this study was to compare clinical characteristics and prognosis according to PR expression levels in invasive breast cancer patients. Methods: A prospective cohort study was conducted to identify YBC patients with invasive carcinoma diagnosed at an age of less than 40 years old between 2013 and 2018. Clinicopathologic features and prognosis of ER-positive and human epidermal growth factor receptor 2 (HER2)-negative patients were investigated. Patients were stratified into strong PR (PR-positive cell proportion > 10%), low PR (PR-positive cell proportion = 1~10%), and PR-negative (PR-positive cell proportion < 1%). Results: Among 458 patients enrolled, 386 (84.3%), 26 (5.7%), and 46 (10.0%) were categorized into strong PR, low PR, and PR-negative groups, respectively. The median follow-up duration was 58.6 months. Compared with the strong PR group, low PR and PR-negative groups were more likely to have high Ki-67 and a high nuclear grade. Low R and PR-negative groups had significantly worse disease-free survival (DFS) and distant metastasis-free survival (DMFS) than the strong PR group (*p* = 0.0033, *p* = 0007). Low PR group had an even higher risk of distant metastasis than PR-negative patients. Low PR patients and PR-negative had significantly lower overall survival (OS) rates than strong PR. Conclusion: Low PR might be a prognostic factor of ER-positive/HER2-negative in YBC.

## 1. Introduction

The status of hormone receptors (HR) including estrogen receptor (ER) and progesterone receptor (PR) is known to be correlated with response to endocrine therapy in breast cancer [1]. The type of breast cancer, the status of its estrogen receptor site, and how it might respond to therapy are crucial factors affecting treatment success, ultimately affecting outcomes and overall survival (OS) [2]. An accurate diagnosis of HR positivity and treatment plans are also crucial for patient survival and longevity. PR expression is known to be inversely associated with recurrence score and other characteristics such as mitosis and luminal B subtype [3]. PR is a useful marker of functional ER. The expression of PR approximates ER activity. Hormonal cross-talk of ER with PR-A is a fundamental mechanism that promotes the invasiveness and metastatic potential of HR-positive breast cancers by suppressing the regulation of critical microRNAs by estrogen [4]. Therefore, high expression of PR is more frequently observed in cancers with a better baseline prognosis (such as luminal A subtype) than in cancers with a poor baseline prognosis (such as luminal B subtype) [5].

Several studies have suggested that single HR-positive tumors without human epidermal growth factor receptor 2 (HER2) overexpression have aggressive clinical features and poorer survival than double HR-positive without HER2 overexpression tumors and triple-negative breast cancer [1]. Patients with double HR-positive cancers who are simultaneously ER-positive and PR positive have considerably longer median OS than patients with single HR-positive tumors [6]. Patients with PR-positive cancers also have a longer median OS than patients with PR-negative tumors in ER-positive tumors [7]. Previous studies have shown that PR is sufficient to drive tumor growth and metastasis in ER-signaling ablated tumor cells. Therefore, abrogating PR expression might be a therapeutic strategy in further treatment options [8]. Even we expect that PR status would be an independent predictive factor for obtaining benefits from an adjuvant endocrine therapy [9].

Although ER expression levels affect the prognosis of breast cancer, studies about PR expression levels are insufficient, especially in young breast cancer (YBC) [10]. There is a knowledge gap between single HR-positive and double HR-positive, especially according to the PR expression level in YBC patients [11]. Furthermore, the characteristics and prognosis of patients with low PR cancer are unknown. To investigate the prognosis of patients according to PR expression level, breast cancer patients who were ER-positive and HER2-negative were compared after stratifying them according to PR level.

## 2. Methods

### 2.1. Patient Selection

A prospective cohort study was conducted to identify YBC patients. YBC patients were selected from the clinical database of the Breast Cancer Center at Samsung Medical Center (SMC), Republic of Korea, between January 2013 and December 2018. YBC patients diagnosed at an age less than 40 years old, during pregnancy, or within one year postpartum were included. Breast cancer patients with ipsilateral invasive carcinoma and biologic factors who were ER-positive and HER2-negative were selected. Patients who were diagnosed with bilateral tumors, metaplastic, mucinous, or mixed carcinoma were excluded (Figure 1).

### 2.2. PR Stratification/Biologic Factors Definitions

PR expression level was determined based on positive cell proportion after immunohistochemical cell (IHC) staining. A low level of PR expression was defined as 1~10% cell staining. Patients were divided into three groups: strong PR (group I, PR-positive cell proportion > 10%), low PR (group II, PR-positive cell proportion = 1~10%), and PR-negative (group III, PR-positive cell proportion < 1%).

HER2 positivity was defined as an intensity of 3+ by IHC. A score of 2+ was interpreted as equivocal. A staining test result of 0/1+ was considered negative. Silver in situ hybridization (SISH) or fluorescence in situ hybridization (FISH) was used for equivocal staining. Results were positive for HER2 amplification when the ratio of HER2 to CEP17 was >2.0. For the Ki-67 level, results were considered positive based on the identification of the following criteria in at least one core and when 20.0% of cells showed staining. We reviewed the clinicopathologic characteristics of patients, and biological factors such as Ki-67 were added to ER, PR, and HER2. In the neoadjuvant chemotherapy cases, all the biological factors, especially ER and PR expression levels, were based on core needle biopsy samples. The pathologic tumor stage was assessed according to the 7th American Joint Committee on Cancer.

### 2.3. Statistics

Using the chi-square test and Fisher’s exact test, differences in frequencies of clinicopathological variables and subtypes were statistically analyzed. Disease-free survival (DFS) was defined as the time from surgery to the date of documentation of relapse, including locoregional recurrence and/or distant metastasis. The number of months from surgery to death was defined as OS. Distant metastasis-free survival (DMFS) was defined as the time from surgery to the date of documentation of any evidence of distant metastasis. Survival curves were constructed using the Kaplan–Meier method. Hazard ratios were estimated using a Cox regression for DFS/OS in univariate and multivariate analyses. Statistical significance was defined as *p* < 0.05. All statistical analyses were executed using SAS version 9.4 (SAS Institute, Cary, NC, USA).

## 3. Results

A total of 1087 young breast cancer patients were identified. Of them, 970 patients were selected for this study after excluding patients who were diagnosed with bilateral tumors, metaplastic, mucinous, or mixed carcinoma. We selected 458 patients with ipsilateral invasive carcinoma who were ER-positive and HER2-negative.

### 3.1. Clinicopathologic Characteristics

The basic clinicopathologic characteristics were summarized in Table 1. The median follow-up duration for 458 patients included in this analysis was 58.6 months. Their median age was 36.5 years (range, 20~43 years). There were 386 (84.3%), 26 (5.7%), and 46 (10.0%) strong PR, low PR, and PR-negative ER+/HER2− YBC patients, respectively. There were 433 (94.5%) cases diagnosed as invasive ductal carcinoma and 25 (5.5%) cases diagnosed as invasive lobular carcinoma. Overall, ER+/HER2− YBC patients had stage T1 and N0 the most. Most patients were at stage I (40.6%), or II (40.8%) (*p* = 0.027). The additional pathologic staging analysis was disclosed with a supplement. Patients who received neoadjuvant chemotherapy and those who did not were separately analyzed (Appendix A). Low PR (42.3%) and PR-negative (63.0%) patients had higher nuclear grades than strong PR (13.0%) patients (*p* < 0.0001). Compared with the strong PR group (26.7%), low PR (65.4%) and PR-negative (78.3%) groups were more likely to have high Ki-67 (*p* < 0.0001). In terms of lympho-vascular invasion of invasive carcinoma, low PR patients had more presence of lympho-vascular invasion than the absence of lympho-vascular invasion (57.7% vs. 42.3%, *p* = 0.0003). Otherwise, strong PR patients (46.9% vs. 53.1%) and PR-negative patients (17.4% vs. 82.6%) had more absence of lympho-vascular than presence of lympho-vascular invasion (*p* = 0.0003). All ER+/HER2− patients had more single carcinoma lesions than multiple lesions (*p* = 0.014). There was no difference in BRCA1 or BRCA2 mutation (BRCA1 *p* = 0.0531, BRCA2 *p* = 0.3640).

Approximately 97.2% of ER+/HER2− YBC patients received endocrine therapy, 42.4% received adjuvant chemotherapy, and 22.5% received neoadjuvant chemotherapy (*p* < 0.0001) (Table 1). There was no significant difference in the rate of patients who received adjuvant chemotherapy (*p* = 0.7450) or radiation therapy (*p* = 0.2654) among the three groups. However, the low PR group received chemotherapy more than the PR-negative group (96.2% vs. 91.3%, *p* < 0.0001). Both low PR and PR-negative groups received chemotherapy more than the strong PR group (58.8%). PR-negative patients received less endocrine therapy than low PR and strong PR patients (80.4% vs. 100% and 99.0%, *p* < 0.0001). Regarding the proportion of those receiving neoadjuvant chemotherapy, low PR (50% vs. 50%, *p* < 0.0001) and PR-negative (47.8% vs. 52.2%, *p* < 0.0001) patients received neoadjuvant chemotherapy more than strong PR patients (17.6% vs. 82.4%, *p* < 0.0001).

### 3.2. Oncologic Outcomes

With the Kaplan–Meier method, the DFS, DMFS, and OS rate graph of low PR patients was located between that of strong PR and PR-negative patients or below strong PR and PR-negative patients (Figure 2). In terms of recurrence of ER+/HER2− YBC (*p* = 0.0033), the 5-year DFS rate of low PR patients and PR-negative patients were 76.4% and 74.8%, respectively, significantly lower than that of strong PR patients (87.5%). In terms of distant metastasis of ER+/HER2− YBC, low PR patients (76.3%) showed lower DMFS rates than strong PR patients (92.5%) and PR-negative patients (77.4%) (*p* = 0.0007). The last prognostic factor in this study was death (*p* < 0.0001). PR-negative patients showed a lower 5-year OS rate (81.4%) than strong PR patients (97.4%) and low PR patients (92.2%) (*p* < 0.0001). Low PR patients had lower OS rates than strong PR patients (*p* < 0.0001).

With univariate analysis by Cox regression, low PR and PR-negative patients had increased risks of recurrence, death, and distant metastasis compared with strong PR patients (Table 2). Specifically, compared with strong PR patients, low PR patients had increased risks of recurrence, death, and distant metastasis with a hazard ratio of 2.321 (*p* = 0.053, 95% CI: 0.991–5.439) for DFS, 2.905 (*p* = 0.337, 95% CI: 0.512–16.49) for OS, and 2.961 (*p* = 0.017, 95% CI: 11.171–7.486) for DMFS. PR-negative patients had increased risks of recurrence, death, and distant metastasis with a hazard ratio of 2.398 (*p* = 0.013, 95% CI: 1.17–4.918) for DFS, 7.709 (*p* < 0.0001, 95% CI: 2.660–22.342) for OS, and 2.887 (*p* < 0.0001, 95% CI: 1.297–6.43) for DMFS compared with strong PR patients. With multivariate analysis by Cox regression, only overall survival had significant results with an overall *p*-value of 0.005. Low PR patients only had an increased risk of death with a hazard ratio of 2.270 (*p* = 0.566, 95% CI: 0.319–9.835) compared with strong PR patients. However, PR-negative patients had a hazard ratio of 6.257 (*p* = 0.002, 95% CI: 1.818–21.479).

## 4. Discussion

ER-positive/HER2-negative YBC patients in low PR and PR-negative groups had similar clinicopathological characteristics, such as higher nuclear grade and higher Ki-67, than those in the strong PR group, with low PR patients having more lympho-vascular invasion than PR-negative patients. ER-positive/HER2-negative YBC patients in low PR and PR-negative groups were associated with an increased risk of recurrence, distant metastasis, and death compared with those in the strong PR group. Low PR patients had even lower distant metastasis-free survival rates than PR-negative patients. ER-positive/HER2-negative YBC patients with low PR had an increased risk of death with a hazard ratio of 2.270 compared with strong PR patients in multivariate analysis with a significant *p*-value.

Since 2020, guidelines by ASCO-CAP have mandated that breast cancer specimens with ≥1% positively staining cells by immunohistochemistry should be considered ER-positive, the concept of a subclass of low ER-positive (1–10%) has emerged. Some studies have examined the clinical characteristics and prognosis of low ER-positive patients and found that low ER-positive breast cancers behave like HR-negative tumors [12,13,14]. Several studies have recently revealed the importance of subdividing HR-positive as single hormone-positive receptors or low PR patients [11,15,16]. Phenotypes of a single HR can determine differences in patient demographics, tumor characteristics, and prognostic outcomes, such as unfavorable characteristics and poorer survival than ER-positive/PR-positive subtypes [11,15,16]. ER-positive/PR-negative tumors have a higher grade than double HR-positive tumors, although they have a lower grade than double HR-negative tumors [17]. Therefore, different strategies might be required for patients with single HR-positive tumors to ensure optimal treatment and maximum benefits from therapies [18]. Endocrine therapy has a significant benefit for patients whose tumors express high HR levels and a favorable tendency for patients with tumors expressing low HR levels [19]. RT-PCR followed by IHC has also been suggested because low ER status is important to ensure prognosis and needed more grounds for appropriate treatment planning [20]. Some guidelines have stated that for patients with HR low positive breast cancer, endocrine therapy alone might be insufficient, and additional neoadjuvant or adjuvant chemotherapy should be considered [21,22,23,24].

Out of HR low positive breast cancer concept, studies about low PR breast cancer are rare. PR status should be considered when discussing relative risk reductions expected from endocrine treatment with individual patients [9]. TP53 mutation rate and median SUVs (standardized uptake values, from 18F-FDG PET) are significantly higher in low PR tumors than in high PR cancers [25]. Multivariable analysis has disclosed that SUV and age remained predictive variables associated with low PR expression in ER-positive and HER2-negative breast cancer [26]. Reverse transcription polymerase chain reaction (RT-PCR)-based tests such as 21-gene assay and oncotype DX have revealed that expression levels of ER, PR, and HER2 mRNA are associated with cancer progression and adverse patient outcomes. Individual ER, PR, and HER2 expression scores with recurrence score^®^ (RS^TM^) generated from a validated algorithm that compares expression levels of 16 cancer-related genes to the expression of five housekeeping control reference genes have been used to predict the risk of disease recurrence within 10 years after treatment [27]. Low PR expression has been found to be associated with a high RS and other clinicopathologic features such as high tumor grade, infiltrating ductal histology, and high HER-2 expression traditionally [28,29]. PR negativity, luminal B type, and mitosis are strongly correlated with a higher RS [3].

Our study suggests that not only low ER concepts, but also low PR concepts are important to understand HR-positive breast cancer characteristics. Each classification of HR-positive breast cancer patients needs to be considered as having different physiology because it shows a different prognosis which affects our decision of different treatment plans. Because low PR patients had worse prognoses, like those with PR-negative breast cancer, patients who are HR-positive with low PR must be considered as different from double HR-positive patients. In particular, HR status is necessary for YBC including most pre-menopausal patients. Several studies have emphasized that ER-positive breast cancer is more aggressive in pre-menopausal patients than in post-menopausal patients [8]. YBC patients have a higher incidence of negative clinicopathologic features such as higher histologic grade, more lymph node positivity, lower ER positivity, and higher rates of Her2/neu overexpression [30]. Therefore, this study only included patients younger than 40 years old to have meaningful and significant results. YBC patients have a worse prognosis and higher RS than patients who are older. Thus, HR positivity in YBC should be considered important. Those who are HR-positive are not safe anymore if PR positivity is low in YBC patients. Among HR-positive and HER2-negative YBC patients, we additionally stratified ER-positive patients as strong ER and low ER and compared prognostic outcomes after stratifying patients according to PR expression level. For strong ER patients, disease-free survival rate and distant metastasis-free survival rate were the lowest for low PR patients, even lower than that of PR-negative patients (DFS: 68.0% vs. 76.0%, *p* = 0.009; DMFS: 72.0% vs. 84.6%, *p* = 0.000). Both low ER and low PR patients were very rare. There was only one such patient in our study. Therefore, there was no statistically meaningful result for these subtypes.

This study has several limitations. First, considering that 22% of ER+/HER2− YBC patients received neoadjuvant chemotherapy, pathological stage interpretation was incomplete. This is a fundamental problem of neoadjuvant treatment since neoadjuvant chemotherapy has been started for breast cancer patients. Nevertheless, we analyzed pathologic staging with patients who received neoadjuvant chemotherapy and those who did not separately for comprehensive staging data. With further studies about neoadjuvant chemotherapy and its oncological outcomes, an optimal statistical stage standardization whether pathologic or clinical stage would be discussed soon. Second, the number of patients was relatively small. There were only 458 patients with ER+/HER2− YBC. The analysis was conducted only with YBC patients which is practically a small proportion of total breast cancer patients. According to an overview of YBC in 2017, women diagnosed in aged <40 years was accounting for 4~5% of all women diagnosed with breast cancer [31]. The proportion was so small that further stratification and analyses by age with those YBC groups were practically difficult. Furthermore, the proportion of PR-positive breast cancer is lower than ER-positive breast cancers basically, and 20% of ER-positive breast cancers were PR-negative in SEER breast cancer registries [16,32]. No such pattern has been seen with PR expression proportion, but the PR-positive rate remains constant in all age groups. Moreover, it is an unchangeable fact that the proportion of PR-negative and low PR groups is smaller than the strong PR group. In our study, the number of low ER and low PR patients for subgroup analysis according to the neoadjuvant chemotherapy option was only one. However, the data were enough to obtain statistically significant results. We should not ignore the fact that PR expression cannot predict the benefit of endocrine efficacy in ER-positive patients because the profile of gene expression is not completely equal to clinical IHC-based HR status, although they are closely related to each other. A nationwide multicenter study is needed to overcome these limitations for abundant subgroup analysis such as analysis for low ER and low PR patients. Efficacy of endocrine therapy also needs to be determined for patients with HR-positive breast cancer according to HR expression level.

## 5. Conclusions

For ER-positive/HER2-negative YBC, low PR patients show aggressive clinicopathologic characteristics and significantly worse DFS, DMFS, and OS than strong PR patients. They even show worse DMFS than PR-negative patients. Therefore, low PR might be a prognostic factor for a poor outcome of ER-positive/HER2-negative in YBC.

## Figures and Tables

**Figure 1 cancers-15-03435-f001:**
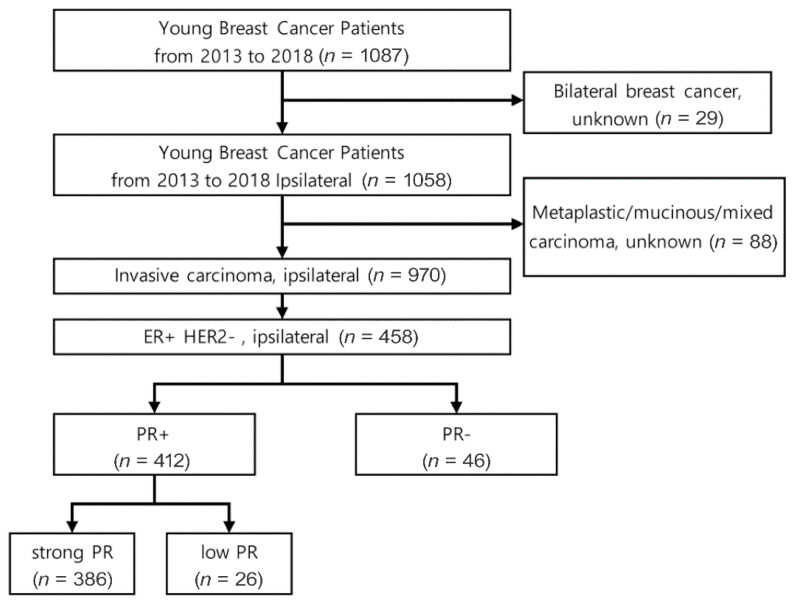
Flowchart of patient selection.

**Figure 2 cancers-15-03435-f002:**
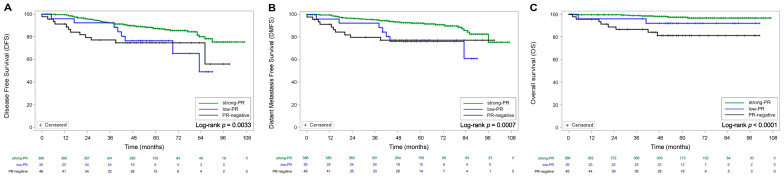
(**A**) Disease-free survival (DFS) rate, (**B**) distant metastasis-free survival (DMFS) rate, and (**C**) overall survival (OS) rate of young breast cancer patients with ER+/HER2−.

**Table 1 cancers-15-03435-t001:** Clinicopathologic characteristics of YBC patients with ER+/HER2−.

ER+HER2− Characteristics	Strong PR (*n* = 386, 84.3%)	Low PR (*n* = 26, 5.7%)	PR-Negative (*n* = 46, 10.0%)	*p* Value
Age, median (range)	37 (20–43)	35.5 (28–40)	34.5 (25–40)	0.0121
Dx				0.1889
IDC	362 (93.8)	25 (96.2)	46 (100)	
ILC	24 (6.2)	1 (3.8)	0 (0)	
pT				0.0566
pCR	4 (1.0)	2 (7.7)	3 (6.5)	
T1	210 (54.4)	12 (46.2)	26 (56.5)	
T2	142 (36.8)	9 (34.6)	15 (32.6)	
T3	30 (7.8)	3 (11.5)	2 (4.3)	
pN				0.3810
N0	219 (56.7)	12 (46.2)	31 (67.4)	
N1	113 (29.3)	8 (30.8)	11 (23.9)	
N2	40 (10.4)	3 (11.5)	3 (6.5)	
N3	14 (3.6)	3 (11.5)	1 (2.2)	
Stage				0.0270
pCR	4 (1.0)	2 (7.7)	3 (6.5)	
I	159 (41.2)	7 (26.9)	20 (43.5)	
II	159 (41.2)	10 (38.5)	18 (39.1)	
III	64 (16.6)	7 (26.9)	5 (10.9)	
Nuclear Grade				<0.0001
low	38 (9.8)	0 (0.0)	5 (10.9)	
intermediate	298 (77.2)	15 (57.7)	12 (26.1)	
high	50 (13.0)	11 (42.3)	29 (63.0)	
Ki-67				<0.0001
≤20.0%	283 (73.3)	9 (34.6)	10 (21.7)	
>20.0%	103 (26.7)	17 (65.4)	36 (78.3)	
LVI				0.0003
yes	181 (46.9)	15 (57.7)	8 (17.4)	
no	205 (53.1)	11 (42.3)	38 (82.6)	
Multiplicity				0.0137
yes	131 (33.9)	7 (26.9)	6 (13.0)	
no	255 (66.1)	19 (73.1)	40 (87.0)	
BRCA1 Mutation				0.0531
not detected	367 (95.1)	24 (92.3)	40 (87.0)	
equivocal	17 (4.4)	1 (3.8)	5 (10.9)	
detected	2 (0.5)	1 (3.8)	1 (2.2)	
BRCA2 Mutation				0.3640
not detected	339 (87.8)	23 (88.5)	41 (89.1)	
equivocal	29 (7.5)	0 (0.0)	3 (6.5)	
detected	18 (4.7)	3 (11.5)	2 (4.3)	
Breast Surgery				0.1609
BCS	216 (56.0)	13 (50.0)	32 (69.6)	
TM	170 (44.0)	13 (50.0)	14 (30.4)	
Axillary Surgery				0.1706
SLNB	253 (65.5)	13 (50.0)	33 (71.7)	
ALND	133 (34.5)	13 (50.0)	13 (28.3)	
Adjuvant Radiation therapy				0.2654
Yes	294 (76.2)	20 (76.9)	39 (84.8)	
No	90 (23.3)	6 (23.1)	6 (13.0)	
unknown	2 (0.5)	0 (0.0)	1 (2.2)	
Adjuvant Endocrine Therapy				<0.0001
Yes	382 (99.0)	26 (100)	37 (80.4)	
No	2 (0.5)	0 (0.0)	8 (17.4)	
unknown	2(0.5)	0 (0.0)	1 (2.2)	
NAC				<0.0001
Yes	68 (17.6)	13 (50.0)	22 (47.8)	
No	318 (82.4)	13 (50.0)	24 (52.2)	
Adjuvant Chemotherapy				0.7450
Yes	160 (41.5)	12 (46.2)	22 (47.8)	
No	224 (58.0)	14 (53.8)	24 (52.2)	
unknown	2 (0.5)	0 (0.0)	0 (0.0)	
Chemotherapy				<0.0001
Yes	227 (58.8)	25 (96.2)	42 (91.3)	
No	159 (41.2)	1 (3.8)	4 (8.7)	

**Table 2 cancers-15-03435-t002:** Uni- and multivariate analysis of disease-free survival (DFS), distant metastasis-free survival (DMFS), and overall survival (OS) of young breast cancer patients with ER+/HER2−.

			Hazard Ratio	*p* Value	Overall *p* Value	95% CI	
						Lower	Upper
DFS	univariable	strong PR vs. low PR	2.321	0.0533	2.8870	0.991	5.439
		strong PR vs. PR-negative	2.398	0.0127		1.170	4.918
	multivariable	strong PR vs. low PR	1.017	1.0000	0.9832	0.381	2.353
		strong PR vs. PR-negative	0.933	1.0000		0.343	2.193
DMFS	univariable	strong PR vs. low PR	2.961	0.0174	0.0013	1.171	7.486
		strong PR vs. PR-negative	2.887	<0.0001		1.297	6.430
	multivariable	strong PR vs. low PR	1.524	0.6480	0.4800	0.319	9.835
		strong PR vs. PR-negative	1.537	0.7465		1.818	21.479
OS	univariable	strong PR vs. low PR	2.905	0.3373	<0.0001	0.512	16.49
		strong PR vs. PR-negative	7.709	<0.0001		2.660	22.342
	multivariable	strong PR vs. low PR	2.270	0.5661	0.0051	0.319	9.835
		strong PR vs. PR-negative	6.257	0.0024		1.818	21.479

## Data Availability

The datasets generated and/or analyzed during the current study are available from the corresponding author upon reasonable request.

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
