# Peer review of "Progesterone Receptor Expression Level Predicts Prognosis of Estrogen Receptor-Positive/HER2-Negative Young Breast Cancer: A Single-Center Prospective Cohort Study"

_cancers, 2023, doi:10.3390/cancers15133435_

Round 1

Reviewer 1 Report

The authors analyzed the expression of progesterone receptor in young breast cancer patients being less than 40 years of age in order to find out whether the expression of PR modifies the clinical course of breast cancer. The study was done on 458 breast cancer patients (ER+, HER-, ipsilateral).

The statistical analysis was the state of art. The authors had detailed information about the clinicopathologic features of the breast cancer patients under study. They got interesting results that low progesterone receptor expression is connected with worse clinical course. They are also critical about the No of patients in the study. The manuscript is interesting.

Author Response

We sincerely appreciate your comments and explanation for our study. We consider our study conducted only with young breast cancer cohort, which was only including less then 40 years of age, is significant and meaningful because hormone receptor positivity is important prognostic factors in particularly YBC patients. Thank you again for your complementation of statistically significant result of clinicopathologic features.

Reviewer 2 Report

The manuscript arises from a statistical analysis of medical records of 458 ER+/HER2- breast cancer patients. There are many concerns about the analysis. PR low and PR negative groups are very small compared with PR-strong group. The three groups have statistically significant differences, but it is not clear to understand (it is not included in tables or in the figure) whether the difference is between PR-positive and PR-negative (or low) group or between PR-negative and PR-low group. The age is different among the groups but it is not considered in the following analyses. 

Author Response

We appreciate your comments for our study, we concurred in the suggestion of necessity of larger cohort for more satisfactory statistically. The analysis was only with young breast cancer patients which is practically small proportion of total breast cancer patients. According to an overview of young breast cancer in 2017, women diagnosed in aged <40 years is accounting for 4–5% of all women diagnosed with breast cancer1. The proportion is so small that further stratification by age and analyses with those young breast group is difficult practically. We totally agreed your opinion and the fact that further research would be necessary with larger cohort and with detailed age stratification.

Second, the proportion of PR positive breast cancer is lower than ER positive breast cancers with various references2. Furthermore, in SEER breast cancer registries, 20% of ER positive breast cancers were PR negative3. No such pattern has been seen with PR expression proportion, and the PR positive rate remains constant in all age groups. It is unchangeable proportion which PR negative and low-PR groups are smaller than strong-PR group. Ultimately, large cohort is needed for analysis with more low-PR patients.

In terms of the comparison issue of outcome between three group, PR-negative and low-PR groups had lower DMFS, DFS and OS than strong PR group. Only with distant metastasis, low-PR group had worse DMFS than even PR-negative group. We indicated the percentage of 5-year prognostic outcomes in the text of result.

References

  1. Anastasiadi Z, Lianos GD, Ignatiadou E, Harissis HV, Mitsis M. Breast cancer in young women: an overview. Updates Surg. 2017 Sep;69(3):313-317. doi: 10.1007/s13304-017-0424-1. Epub 2017 Mar 4. PMID: 28260181.
  2. Dunnwald LK, Rossing MA, Li CI. Hormone receptor status, tumor characteristics, and prognosis: a prospective cohort of breast cancer patients. Breast Cancer Res.9, R6 (2007)
  3. Chow LW, Ho P. Hormonal receptor determination of 1,052 Chinese breast cancers.  Surg. Oncol.75, 172–175 (2000)

Reviewer 3 Report

In your interesting study on young breast cancer patients (estrogen receptor positive and human epidermal growth factor 2 negative) examination of low-progesterone receptor (PR) and PR-negative groups had significantly worse disease-free survival (DFS) and distant metastasis free survival (DMFS) than the strong-PR group. Also, both the low-PR and PR-negative groups received chemotherapy more than the strong-PR group. Your results suggest that low and negative PR groups may be a prognostic factor for a poor outcome in these groups. Further studies are encouraged by your group to develop this association.

Please rewrite this second from last sentence in abstract: “There was a significant worse in overall survival (OS), low-PR patients had lower OS rate than strong-PR, although both 30 were PR-positive groups. “

Also please rewrite these sentences:

Line 204: “Multivariable analysis has revealed that high SUV and age greater than 70?? remained independent variables but are associated with low PR expression in ER-positive and HER2-negative breast cancer.

Line 206: Reverse-transcription

polymerase chain reaction (RT-PCR)-based tests such as 21-gene assay and oncotype DX have revealed that expression levels of ER, PR, and HER2 mRNA are associated with cancer progression and adverse patient outcomes.

Please rewrite this second from last sentence in abstract: “There was a significant worse in overall survival (OS), low-PR patients had lower OS rate than strong-PR, although both 30 were PR-positive groups. “

Also please rewrite these sentences:

Line 204: “Multivariable analysis has revealed that high SUV and age greater than 70?? remained independent variables but are associated with low PR expression in ER-positive and HER2-negative breast cancer.

Line 206: Reverse-transcription

polymerase chain reaction (RT-PCR)-based tests such as 21-gene assay and oncotype DX have revealed that expression levels of ER, PR, and HER2 mRNA are associated with cancer progression and adverse patient outcomes.

Author Response

We sincerely appreciate your particular and detailed comments for our study. Our study suggested that PR stratification has significant meaning, and low PR expression relates to worse clinical and oncological outcomes.

We affirmatively considered your suggestions and revised sentences for more understandable and articulate explanation for the study. Revised sentences were listed below.

Furthermore, we attached a certificate of editing for this manuscript by professional English editors.

* second from last sentence in abstract

-> Low-PR patients and PR-negative had significantly lower overall survival (OS) rate than strong-PR.

* Line 204

-> Multivariable analysis has disclosed that SUV and age remained predictive factors associated with low PR expression in ER-positive and HER2-negative breast cancer.

* Line 206

-> Reverse-transcription polymerase chain reaction (RT-PCR)-based tests such as 21-gene assay and oncotype DX have revealed that expression levels of ER, PR, and HER2 mRNA are associated with cancer progression and adverse patient outcomes.

Reviewer 4 Report

Major revision

1.      At the pT and pN stage in Table 1, patients who received neoadjuvant chemotherapy and those who did not were mixed and analyzed. These case may confuse the readers. Whether it is shown separately or compared by clinical stage not pathologic stage, the same standard should be used.

2.      In the neoadjuvant chemotherapy cases, it is also necessary to clearly define whether the low PR or PR-negative is based on core biopsy sample or the residual lesion after neoadjuvant treatment.

3.      Results and Discussion;

The main outcome of this study is that PR-low or -negative is a significant factor affecting survivals. However, PR-low or -negative generally has a high correlation with ER low or –negative status. The correlation between PR expression and ER expression was not addressed in this study. Further analysis is needed on this.

4.      If ER low expression and PR low expression were indeed positively correlated in this study, it would simply be a comparison of HR+/HER2- subtype and the actual TNBC subtype. Then, how unique is this study compared to previous studies?

Minor revision

1.      Figure 2(b) “Figure 2. (A) Disease Free Survival (DFS) rate, (B) Distant Meta Free Survival (DMFS) rate and (C) 153 Overall survival (OS) rate of young breast cancer patients with ER+/HER2-.”

è Abbreviations not defined (meta-> metastasis)

Minor editing of English language required

Author Response

We sincerely appreciate that your particular and detailed comments for our study. Here are replies to revision points. 

Major revision point 1.

We affirmatively considered your suggestions and added the description about how we managed the data with neoadjuvant chemotherapy cases to clarify (Method, Line 120). In the neoadjuvant chemotherapy cases, all the biologic factors, especially ER and PR expression level, were based on core needle biopsy sample.

Major revision point 2.

We analyzed and added a supplementary material about additional pathologic staging analysis (Result, Line 147). Pathologic staging with patients who received neoadjuvant chemotherapy and those who did not were separately analyzed in the table (table S1).

Major revision point 3 and 4.

In terms of the comment for result and discussion about correlation between PR expression and ER expression, as mentioned in limitation, simultaneously low ER and low PR patient was only one which is preposterously insufficient data to analyze. Basically, because this study was conducted in cohort with only ER-positive and HER2-negative young breast cancer patients, analysis for PR stratification with ER stratification (including ER-negative patients) cannot be validated unfortunately.

Minor revision point 1.

We appreciated again for your meticulous review and comment about the error with undefined abbreviation. We changed and revised the details in figure and main document.

Round 2

Reviewer 2 Report

Please, add the response to the reviewer's concern in the text of the manuscript.

Author Response

We deeply appreciate again for your comment. We revised the manuscript according to the comments and added the point of reviewers' concerns additionally (Discussion, Line 256-258, Line 261-271). The revised manuscript was attached. 

Reviewer 4 Report

The parts that were insufficient for publication have been corrected.

Author Response

We deeply appreciate for the review and comments that made us consider various point of view. We revised discussion point considering your comments and suggestions.